# Green Emissive Copper(I) Coordination Polymer Supported by the Diethylpyridylphosphine Ligand as a Luminescent Sensor for Overheating Processes

**DOI:** 10.3390/molecules28020706

**Published:** 2023-01-10

**Authors:** Kamila R. Enikeeva, Aliia V. Shamsieva, Anna G. Strelnik, Robert R. Fayzullin, Dmitry V. Zakharychev, Ilya E. Kolesnikov, Irina R. Dayanova, Tatiana P. Gerasimova, Igor D. Strelnik, Elvira I. Musina, Andrey A. Karasik, Oleg G. Sinyashin

**Affiliations:** 1Arbuzov Institute of Organic and Physical Chemistry, FRC Kazan Scientific Center, Russian Academy of Sciences, 8 Arbuzov Street, 420088 Kazan, Russia; 2Centre for Optical and Laser Materials Research, Research Park of Saint Petersburg State University, 5 Ulianovskaya Street, 198504 Saint Petersburg, Russia

**Keywords:** Cu_4_I_4_ clusters, P,N-ligands, copper(I) complexes, phosphines, luminescent complexes

## Abstract

Tertiary diethylpyridylphosphine was synthesized by the reaction of pyridylphosphine with bromoethane in a suberbasic medium. The reaction of phosphine with the copper(I) iodide led to the formation of a copper(I) coordination polymer, which, according to the X-ray diffraction data, has an intermediate structure with a copper-halide core between the octahedral and stairstep geometries of the Cu_4_I_4_ clusters. The obtained coordination polymer exhibits a green emission in the solid state, which is caused by the ^3^(M+X)LCT transitions. The heating up of the copper(I) coordination polymer to 138.5 °C results in its monomerization and the formation of a new solid-state phase. The new phase exhibits a red emission, with the emission band maximum at 725 nm. According to the experimental data and quantum chemical computations, it was concluded that depolymerization probably leads to a complex that is formed with the octahedral structure of the copper-halide core. The resulting solid-state phase can be backward-converted to the polymer phase via recrystallization from the acetone or DMF. Therefore, the obtained coordination polymer can be considered a sensor or detector for the overheating of processes that should be maintained at temperatures below 138 °C (e.g., engines, boiling liquids, solar heat systems, etc.).

## 1. Introduction

Designing luminescent copper(I) complexes is one of the current fields of interest in modern coordination chemistry; this is strongly motivated by the low cost and toxicity and the high availability of copper, relative to the noble and rare earth metals [1,2,3,4,5,6]. Copper(I) complexes are a large family of coordination compounds, displaying a variety of structures that depend on the denticity and bridging ability of the supporting ligands and counter ions [7,8,9,10,11,12,13,14,15].

Usually, the copper(I) halides form di- or tetranuclear clusters with the μ^2^-, μ^3^- or μ^4^- bridging coordination of halogen ions [9,10]. The tetranuclear copper(I) complexes are presented by clusters with three types of copper-halide cores (Figure 1): cubane-type [8,16,17,18,19,20,21,22,23], octahedral [17,24,25,26,27,28,29,30,31], and stairstep [32,33,34,35,36,37,38].

The complexes with cubane-type clusters are the most studied representatives of the tetranuclear complexes, which display dual-emission behavior with high sensitivity to temperature changes [8,16,21,23,39]. This type of complex has an L_4_Cu_4_X_4_ composition (where L is a P-, N-, S- or other monodentate ligand). The octahedral and stairstep types of the tetranuclear complexes have an L_2_Cu_4_X_4_ composition (where L is a bidentate ligand) [24,25,40]. The division into the octahedral and stairstep classes of complexes is based on the geometry differences of the Cu_4_X_4_ cores, which can be found in the bridging coordination of the iodide ions and Cu^…^Cu distances. The octahedral clusters consist of a square-bipyramid formed by the four Cu(I) ions as the base of a bipyramid bounded by the two μ^4^-iodide ions as the apexes, with two μ^2^-iodide ions and two μ^2^-bidentate ligands lying in the one plane with a Cu_4_-core. The structure of the stairstep-type core is determined by a zigzag chain, alternating the copper(I) and µ^3^-coordinated iodide links. In this type of cluster, the four copper(I) ions form a plane supported by the two μ^3^-iodide ions, two μ^2^-iodide ions, and two μ^2^-bidentate ligands. The μ^3^- or μ^4^-coordination of iodide ions is determined by the distance between two copper(I) ions at the core, which is commonly less than the sum of the van der Waals radii in the octahedral clusters (in the range of ca. 2.6–3.0 Å) and more than 3.0 Å in the stairstep types of clusters [17,25,27,34,36]. 

These differences contribute to the differences in the photophysical properties of the two types of complexes. The octahedral types of complexes often demonstrate dual-band emission in the high-energy (blue or green emission) and low-energy (red emission) ranges of the visible spectra. This phenomenon is attributed to emissions from two excited states, with the ^3^(M+X)LCT (metal–halide to ligand charge transfer) or ^3^CC (cluster-centered) origins of transitions [29,31,40,41]. Those complexes with the stairstep type of Cu_4_X_4_ core are able to emit in the full range of spectra; the energy of the emission depends on the chromophore groups contributing to the ^3^(M+X)LCT transitions of the complexes. To the best of our knowledge, there are no examples of stairstep complexes displaying dual-emission behavior. 

The tetranuclear octahedral and stair-step structural motifs generally require stabilization by the bidentate ligands (e.g., P,P-, N,N- or P,N-donor ligands). The copper(I) clusters with P,N-ligands are one of the most studied forms compared with clusters formed with other ligands [25,28,31]. Recently, we have demonstrated that the dual emission of octahedral clusters can be processed by the two excited states, with various distortions of the Cu_4_I_4_ core, which affects the energy of the ^3^(M+X)LCT transitions [29,42]. Moreover, the utilization of these conformationally restricted ligands allowed isolating clusters with two types of distortions of the Cu_4_I_4_ core, which demonstrates individual green or red emissions. This allowed the researchers to attribute the origin of the emissions of both octahedral clusters to ^3^(M+X)LCT transitions [42].

The rich photophysical properties of the copper(I) complexes with various ligands, including an intriguing dual emission, and the new findings of core distortions in complexes with conformationally restricted P,N-ligands motivated us to synthesize the sterically tiny diethylpyridylphosphine and study its complexation toward copper(I) iodide. Instead of the formation of a Cu_4_I_4_L_2_ complex, a one-dimensional coordination polymer with the Cu_4_I_4_L_2_ structural building unit was obtained. Its structure and photophysical properties are presented in this work.

## 2. Results

The diethylpyridylphosphine **1** was obtained via a two-step reaction, starting with primary phosphine PyPH_2_, according to Figure 1 by the previously reported procedure [43].

The obtained phosphine is characterized by a one-septet signal, which is registered at −10.9 ppm (^2^J_PH_ 14.7 Hz) in the ^31^P NMR spectrum. The position of signals, their multiplicity, and relative intensity in the ^1^H NMR spectrum fully correspond to diethylpyridylphosphine [43].

The reaction of phosphine **1** with copper(I) iodide in the 1:2 metal-to-ligand ratio in dichloromethane (Figure 2) leads to the formation of a yellowish crystalline precipitate, which represents the copper(I) polymer according to the X-ray diffraction (XRD) study.

According to the single-crystal XRD, complex **2** is a double polymeric chain that alternates the copper- and µ^3^-coordinated iodide links (Figure 2). The polymer chain is formed along the shortest axis, 0*a*, of the unit cell (*P*2_1_/*n*). The powder XRD pattern of polycrystalline sample **2** (Figure 3) demonstrates good agreement with the single-crystal XRD data and provides evidence of the homogeneity of the sample that is analyzed. One monomeric unit contains the stairstep centrosymmetric Cu_4_I_4_ core, supported by two diethylpyridylphosphine ligands. The monomeric unit can be described as a tetranuclear L_2_Cu_4_I_4_ cluster (Figure 2). In the monomeric unit of the polymer, all the copper(I) ions exist in the tetrahedral geometry of the ligand environment, with the coordination of P–Cu or N–Cu bonds and three Cu–I bonds, where the iodide ions display bridging coordination.

All the copper(I) cations in the Cu_4_ cluster of the monomeric unit lie on one plane, with the value of the distance between the neighboring copper(I) ions being 2.7693(14) and 3.0662(15) Å. Thus, the centrosymmetric Cu_4_-cluster exists as a parallelogram, with angle values of 63.95(4) and 116.05(4)°. The two molecules of pyridyl(diethyl)phosphines are coordinated to the Cu_4_I_4_ core via the P- and N-donor atoms in a head-to-tail manner. The Cu1 and I2 ions are bound by the iodide and copper(I) ions of the further monomeric units; this connectivity forms the continuous chain of the polymer. Furthermore, the chains in the crystal lie in plains parallel to 0*ac*, and stack along the longest axis, 0*b*, forming the layer-by-layer structure of the crystal with a packing index of 68.2%. 

Notably, the stair-step structure of the Cu_4_I_4_ core can be presented as the distorted octahedral Cu_4_I_4_ core, when seen from the other viewpoint (Figure 2). The differences in the structure of the coordination polymer **2** and the published Cu_4_I_4_ octahedral structures lie in the greater distances between the Cu1 and Cu2 ions (3.066 Å in **2**, versus ca. 2.7–2.9 Å in the octahedral clusters), the μ^3^-coordination mode of the I1 ions, and the deviation of the I2-ions from the Cu_4_ plane in coordination polymer **2**. Possibly, the distortion in the structure of **2** is caused by the polymeric assembly; such an arrangement is possible due to the absence of the steric hindrance of the P,N-ligand. Conversely, some movement inside the Cu_4_I_4_ cluster might enable the formation of an octahedral cluster unit. Moreover, the monomeric unit of **2** turns out to be the average structure between the stairstep and octahedral motifs; therefore, we should take into account in later works that the stairstep and octahedral clusters might be the frontier cases of one structural motif, and, possibly, conversion between clusters can be observed.

The obtained polymer is slightly soluble in dichloromethane, acetone, acetonitrile, and DMF, while the dissolution of complex **2** in the DMSO leads to the oxidation of copper(I) to copper(II). The ^31^P NMR spectra display a sharp signal at −1.4 ppm. The shift of the signal in the ^31^P NMR (Δδ_P_ ca. 10 ppm) spectrum is close to that of the previously studied octahedral complexes [40,42]. The ^1^H NMR indicates a slight broadening of the signals relative to the ligand signals, which is to be expected for the copper(I) complexes due to the shielding of the protons by the transition metal cation. The signals of the pyridyl and ethyl protons of the complex are strongly downfield-shifted relative to those of **1** (Δδ ≈ 0.5–1 ppm), which indicates the preservation of the N(Py)–Cu and P–Cu coordination bonds of the complex in solution.

With the aim of understanding the ability of complex **2** to retain its polymeric structure and establish its approximate stoichiometry in solution, diffusion NMR spectroscopy was used. According to the Stokes–Einstein model, the diffusion coefficient for a molecule is inversely proportional to the hydrodynamic molecular radius [44,45,46,47]. The complexes of smaller size and weight should exhibit different diffusion coefficients (SDCs) compared to the polymeric or oligomeric structures, which possess a larger size and weight. Thus a comparison was made for three samples in the DMF-*d_7_* solution (Table 1): ligand **1**, complex **2**, and the previously synthesized octahedral copper(I) iodide cluster **L_2_Cu_4_I_4_** (L is 1-(pyridine-2-yl)phospholane), acting as a reference compound with a predetermined structure [40].

According to the determined self-diffusion coefficients, it can be concluded that the structures of complex **2** and the reference compound, **L_2_Cu_4_I_4_**, have similar molecular volume; therefore, complex **2** does not retain a polymeric or even oligomeric structure in solution but exists as a molecular compound in solution. According to the shifting of the signals of phosphorus atoms and pyridyl protons, it is possible to suggest that the complex partially retains the structural motif with the P–Cu and N–Cu coordination bonds. The existence of sample **2** as a molecular complex in a solution is also supported by the ESI mass spectra of complex **2**, which indicates the presence of the peaks of the cations with [2L+3Cu+2I]^+^, [2L+2Cu+I]^+^, and [2L+Cu]^+^ composition. The analogous fragmentation seen in the ESI mass spectrometry was observed earlier for the copper(I) complexes with the octahedral Cu_4_I_4_ structure of the copper-halide core [29,42].

Complex **2** or its parts is not emissive in the solution but displays moderate emission in the solid state (Figure 4a). The emission spectrum is represented by a band with a maximum of 505 nm, while the excitation spectrum displays a broad band with a maximum in the region of 356–380 nm. The emission of complex **2** displays a dual exponential luminescence decay, with 0.64 and 2.5 μs lifetimes. The large Stokes shift (ca. 8000 cm^−1^) and the microsecond domain of the lifetimes indicate the phosphorescent origin of the emission. The luminescence quantum yield of complex **2** in the solid state (20%) is rather high in comparison with that of the other representatives of the polymeric copper(I) complexes. 

The heating up of the coordination polymer **2** to 138.5 °C leads to a change in the color of the sample emission to red, with the maximum at 725 nm in the emission spectra, although the cooling-down of the sample to room temperature does not mark a return to green emission conditions. Thus, the heating up of sample **2** leads to the solid-state transformation of the polymer to the solid phase **2a**, which was confirmed by differential scanning calorimetry (DSC) (Figure 5).

According to the DSC, the solid-state transformation of polymer **2** into polymer **2a** occurred at 138.2 °C (endothermic peak) without the decomposition of the sample. Further melting, as a complex endothermic event, was observed at 251.7 °C. The rescan of the sample, obtained after crystallization of the melt, displays the absence of pronounced thermal events until its melting. Thus, for the green-emissive sample **2**, the transformation to another crystalline phase, **2a**, occurs within the DSC experiment, which is accessible upon returning to room-temperature conditions. The crystalline phase, **2a**, which was prepared by heating polymer **2** up to 140 °C and annealing for 15 min, displays the same thermal characteristics as the sample in the rescan experiment. The solid-state phase, **2a**, can be converted back to polymer **2** via recrystallization, from either the DMF or acetone solutions. Therefore, the obtained coordination polymer **2** can be considered to be a sensor and can be applied in the detection of overheating in those processes that should take place at temperatures below 138 °C (e.g., engines, boiling liquids, solar heating systems, etc.).

According to the DSC, NMR, and photophysical experiments, it is possible to suggest that polymer **2** does not retain its polymeric structure during heating to 138.2 °C; instead, it transforms into a molecular complex, **2a**, which displays a red emission. Moreover, our recent studies of the Cu_4_I_4_-octahedral complexes demonstrated that such a type of complex is able to demonstrate both green and red emissions, which are caused by the ^3^(M+X)LCT transition of those structures with different symmetry from the copper-halide core [23,34]. According to the XRD data, the monomeric unit of polymer **2** can be regarded as a distorted octahedral Cu_4_I_4_-complex; consequently, the heating of polymer **2** probably leads to the “concerted” movement of the copper(I) and iodide, which, in turn, could result in the formation of the molecular complex **2a** with an octahedral Cu_4_I_4_ core. 

To evaluate the influence of polymer formation on the compound’s photophysical properties, compound **2** was considered in the same way as the monomeric and dimeric molecules within quantum chemical computations (Figure 6). During the optimization of the monomeric unit, the copper-halide core structure transformed from a stairstep into octahedral geometry; at the same time, the central copper-halide fragment in the dimeric model retained its initial stairstep geometry, whereas the peripheral Cu and I atoms tended to adopt an octahedral geometry (Figure 6).

The predicted UV/Vis spectrum for the monomeric model is slightly blue-shifted, compared to the one predicted for a dimer (Figure 6c), and better matches the experimental curve, suggesting the presence of monomeric units in the solution rather than a polymer structure [48,49]. At the same time, the computations predict a T_1_-S_0_ transition (Figure 7) for the monomer in a lower energy range (464 nm) compared to the dimer (437 nm). 

Therefore, the agreement of the quantum chemical calculations with the UV/vis-absorption spectra, the tendency found in the predicted energies of the T_1_-S_0_ transitions, and the thermochemical behavior, as well as the NMR spectroscopic and mass-spectrometric studies of the solutions of **2,** support our assumption that the stairstep structure of polymer **2** is able to transform into a molecular complex with an octahedral copper(I)-halide core when heated up to 138.2 °C. 

## 3. Materials and Methods

All reactions and purification manipulations were carried out under a dry argon atmosphere, using standard vacuum-line techniques. Commercially available solvents were purified, dried, deoxygenated, and distilled before use. Primary phosphine PyPH_2_ was obtained using the standard method described by Redmore [50]. 

The ^1^H NMR (400.13 MHz) and ^31^P NMR (161.96 MHz) spectra were recorded on a Bruker Avance 400 spectrometer (Karlsruhe, Germany), using the residual solvent as an internal reference for ^1^H (δ = 7.26 in CDCl_3_, and δ = 5.31 ppm in CD_2_Cl_2_) and 85% aqueous solution of H_3_PO_4_ as an external reference for ^31^P. The chemical shifts are reported in ppm and coupling constants (*J*) are reported in Hz. 

The DOSY NMR experiments were performed on a Bruker AVANCE-500 spectrometer (Karlsruhe, Germany). The spectrometer was equipped with a Bruker multinuclear z-gradient inverse probe head that is capable of producing gradients with a strength of 50 G cm^−1^. All experiments were carried out at 303 ± 0.2 K. Chemical shifts (δ) were reported relative to DMF (2.900 ppm for an upfield peak of the CH_3_ group) as an internal standard. To prevent convection in the diffusion NMR spectroscopy experiments, the test sample was loaded into a thin-walled glass 3-millimeter NMR tube and then inserted into the standard 5-millimeter tube usually employed for NMR experiments. The capacity between these tubes was filled up with a solvent DMF-d_7_. After Fourier transformation and baseline correction, the diffusion dimension was processed with the Bruker TopSpin software package (version 3.2). The diffusion constants were calculated by the exponential fitting of the data, belonging to individual columns of the pseudo-2D matrix. Single components have been assumed for the fitting routine. All separated peaks were analyzed and the average values were presented. Hydrodynamic radii (R_H_) were calculated from the self-diffusion coefficient Ds, applying the Stokes–Einstein equation: Ds = kT/6πηR_H_, where k is the Boltzmann constant (1.38 × 10^−23^ JK^−1^), T is the absolute temperature (303 K), and η(DMF, 303 K) = 7.66 × 10^−4^ Pa is the viscosity of the solvent. 

ESI measurements were performed using an AmaZon X ion-trap mass spectrometer (Bremen, Germany) in both positive and negative modes. The mass spectra are given as *m/z* values and relative intensities (I_rel_, %). Chloroform and dimethylformamide were used as solvents for the mass spectrometry measurements. 

Elemental analysis was carried out on a EuroVector-3000 (Pavia, Italy). The determination of the phosphorus and copper content was provided by combustion in an oxygen stream, whereas the content of iodine was found using the Scheniger method.

Differential scanning calorimetry (DSC) was carried out on a Netzsch DSC 204 F1 Phoenix calorimeter (τ-sensor) (Hanau, Germany) using cold-welded aluminum cells in the temperature range of 20–110 °C at a scanning rate of 5 °C/min^−1^. To take sample portions, a Sartorius CPA-2P microbalance was used; the weight of the sample was 1–3 mg.

Powder X-ray diffraction. Powder XRD experiments were performed on a Bruker D8 Advance diffractometer (Karlsruhe, Germany) equipped with a Vario setup and a Vantec linear detector. Cu*K*α_1_ radiation, monochromated by a Johansson monochromator, was used for the test. Experiments were performed at room temperature in the Bragg–Brentano geometry with a planar sample. Powder samples were placed on the surface of a single-crystalline silicon plate.

Single-crystal XRD analysis. An X-ray diffraction study of 2 was performed on a Bruker KAPPA APEX II diffractometer (Karlsruhe, Germany). The data collected were processed using the *APEX*4 software. The structure was solved by the direct method, using the *SHELXT* program [51], and then refined by the full-matrix least-squares method on *F*^2^, using the *SHELXL* program [52]. Non-hydrogen atoms were refined in the anisotropic approximation. The hydrogen atoms were inserted at geometrically calculated positions and were included in the refinement as riding atoms.

Deposition number CCDC 2225852 contains the supplementary crystallographic data for compound 2. These data are provided free of charge by the joint Cambridge Crystallographic Data Centre and the Fachinformationszentrum Karlsruhe Access Structures service www.ccdc.cam.ac.uk/structures (accessed on 27 December 2022).

*Crystallographic data for* 2: C_9_H_14_Cu_2_I_2_NP, colorless prism (0.425 × 0.169 × 0.119 mm^3^), formula weight 548.06 g mol^−1^; monoclinic, *P*2_1_/*n* (No. 14), *a* = 7.7606(6) Å, *b* = 19.4506(17) Å, *c* = 9.6518(8) Å, β = 106.079(3)°, *V* = 1399.9(2) Å^3^, Z = 4, T = 173(2) K, *d*_calc_ = 2.600 g cm^−3^, μ(Mo *K*α) = 7.532 mm^−1^, *F*(000) = 1016; T_max/min_ = 0.1848/0.0712; 31,014 reflections were collected (2.926° ≤ θ ≤ 27.976°, index ranges: −10 ≤ *h* ≤ 10, −25 ≤ *k* ≤ 25, and −11 ≤ *l* ≤ 12), 3312 of which were unique, *R_int_* = 0.0547, *R*_σ_ = 0.0388; completeness to θ of 27.976° 98.3%. The refinement of 138 parameters with no restraints converged to *R*1 = 0.0475 and *wR*2 = 0.1340 for 2781 reflections with *I* > 2σ(*I*) and *R*1 = 0.0590 and *wR*2 = 0.1436 for all data with a goodness-of-fit of *S* = 1.114 and residual electron density ρ_max/min_ = 2.508 and −1.579 e Å^−3^, RMS 0.337; max shift/e.s.d. in the last cycle, 0.001.

Photophysical measurements: UV-Vis spectra were registered at room temperature on a Perkin-Elmer Lambda 35 spectrometer (Rodgau, Germany) with a scan speed of 480 nm min^−1^, using a spectral width of 1 nm. The sample of compound **2** was prepared as a solution in DMF with a concentration of *1.1* 10^−4^ mol·L^−1^ and then placed in 10 mm quartz cells. The photoluminescence properties of the solid-state samples at room temperature were measured on a Fluorolog-3 (Horiba Jobin Yvon, Bensheim, Germany) spectrofluorometer. The powder samples were supported on the quartz glass plates. LED testing (λ_ex_ = 370 nm) was used to carry out lifetime measurements (pulse width 1 ns, and repetition rate 100 kHz). The integration sphere (Quanta–φ, 6 inches) was used to measure the solid-state emission quantum yield. The measurements were carried out with powders, according to the guide provided by the manufacturer (four spectra-based measurements). 

Computational methods: Quantum chemical calculations were performed with the Gaussian 16 (Revision A) [53] suite of programs. The ground/excited-state structures were optimized with the use of the hybrid PBE0 functional [54] model and the Ahlrichs’ triple-ζ def-TZVP AO basis set [55]. In all geometry optimizations, the D3 approach [56], used to describe the London dispersion interactions, together with the Becke–Johnson damping function [57,58,59], were employed as implemented in the Gaussian 16 program. A time-dependent density functional response theory (TDDFT) with the use of CAM-B3LYP [60] (functional) has been employed to compute the vertical excitation energies (i.e., absorption wavelengths) and oscillator strengths for the ground-state optimized geometries in the gas phase, where the 50 lowest singlet excited states were taken into account. The vertical T_1_-S_0_ transition energies were also computed, using the TDDFT approach at the CAM-B3LYP/def-TZVP level, for the triplet geometries optimized by the UPBE0–D3(BJ)/def-TZVP level of theory. The calculated spectra were shifted by −0.24 eV.

Synthesis of ligand **1**: Diethylpyridylphosphine **1** was obtained via the previously reported procedure [43].

To a solution of pyridine-2-ylphosphine (19 mmol) in DMSO (35 mL), a 56% aqueous solution of KOH (57 mmol) was added, and the reaction mixture became red-colored. After 2 h, the reaction mixture was cooled to 5 °C, and a solution of bromoethane (38 mmol) in DMSO (25 mL) was added dropwise over 30 min. The mixture was held at room temperature and stirred overnight. Degassed water (30 mL) was added, then the organic layer was separated via a cannula, and residuary products were extracted with n-hexane (3 × 40 mL) from the aqueous layer. The organic layer was dried over MgSO_4_. The solvent was removed by distillation; the residue was separated by fractional distillation under reduced pressure. Yield 0.72 g (23%). b.p. 47 °C/0.02 mbar. ^1^H NMR (CDCl_3_): *δ*_H_ 8.50 (d, ^3^*J_HH_* = 4.7 Hz, 1H, Py), 7.43 (ddd, ^4^*J_HH_* = 1.9 Hz, ^3^*J_HH_* = ^3^*J_HH_* = 4.8 Hz, 1H, Py), 7.29 (m, 1H, Py), 6.98 (ddd, ^3^*J_HH_* = 6.2 Hz, ^3^*J_HH_* = 4.9 Hz, ^4^*J_HH_* = 2.6 Hz, 1H, Py), 1.73 (q, ^2^*J_HH_* = 15.3 Hz, ^3^*J_HH_* ≈ 7.6 Hz, 2H, -CH_2A_-), 1.61 (q, ^2^*J_HH_* = 15.3 Hz, ^3^*J_HH_* ≈ 7.6 Hz, 2H, -CH_2B_-), 0.9 (ddd, ^3^*J_HH_* ≈ ^3^*J_HH_* ≈ 7.6 Hz, ^3^*J_PH_* ≈ 15.0 Hz, 6H, -CH_3_). ^31^P NMR (CDCl_3_): *δ_P_* −10.9 ppm. 

Synthesis of Complex **2**: To a solution of 1 (0.3 g, 1.79 mmol) in dichloromethane (3 mL), the suspension of copper (I) iodide (0.68 g, 3.59 mmol) in dichloromethane (5 mL) was added. The color of the reaction mixture changed from colorless to yellow, and precipitation of the product was observed. The reaction mixture was stirred for 12 h at room temperature. Afterward, a yellowish precipitate was filtered off, washed with dichloromethane, and dried under a vacuum. Yield: 0.57 g (58%); mp = 136 °C. NMR ^1^H (400 MHz, CD_2_Cl_2_): *δ*_H_ 8.71 (d, ^3^*J*_HH_ = 4.9 Hz, 1H, H-Py), 7.66 (m, 1H, H-Py), 7.64 (dddd, ^3^*J_HH_* ≈ ^3^*J_HH_* ≈ 7.3, ^4^*J_HH_* ≈ ^4^*J_PH_* ≈ 2 Hz, 1H, H-Py), 7.23 (dddd, ^3^*J_HH_* = 4.9, ^3^*J_HH_* = 7.3, ^4^*J_HH_* ≈ ^5^*J_PH_* ≈ 2 Hz, 1H, H-Py), 2.21–2.29 (m, 2H, H-PCH_2_), 2.11–2.20 (m, 2H, PCH_2_), 1.78–1.87 (m, 4H, H-CH_3_). ^31^P NMR (CD_2_Cl_2_) *δ*_P_ −8.4 (br.s). MS (ESI_pos_, *m/z* (*I*_rel_, %), ion): 748 (51) [3L + 2Cu+I]^+^, 584 (63) [2L + 2Cu + I]^+^, 393 (100) [2L + Cu]^+^. Anal. Calcd. for C_26_H_35_N_3_P_3_Cu_2_I_2_: C, 37.00; H, 4.14; Cu, 14.50; I, 28.96; N, 4.79; P, 10.60%. Found: C, 36.95; H, 4.12; Cu, 14.52; I, 29.01; N, 4.74; P, 10.57%. Solid phase **2a** (red emitter). The obtained complex **2** (0.03 g) was heated up to 140 °C and annealed for 15 min to give pure phase **2a**. Reverse synthesis of polymer **2** (green emitter) from **2a**. The obtained complex **2a** (0.05 g) was dissolved in 2 mL of DMF. The solution was dropped onto the glass plate and left for 5 min at room temperature until the crystals formed.

## 4. Conclusions

In summary, it was demonstrated that diethylpyridylphosphine is able to form a copper(I) iodide complex with a polymeric structure and P,N-coordination of the ligand. The new polymer is characterized by the intermediate structure, which can be assigned for both the stairstep and octahedral types of copper halide clusters. Thus, it should be taken into account that stairstep and octahedral clusters might not be individual, rigid cluster types but instead coordination isomers that can transform into one another. The studies of solutions of the polymer in question allowed us to demonstrate that the complex considered does not show the polymeric architecture in solutions, and the solvation of the polymer leads to its monomerization with the formation of the molecular complex. The obtained coordination polymer is the triplet emitter, which displays s green emission with a maximum of 505 nm. The heating of the polymer at 138.5 °C leads to its transformation to a new solid-state phase, which probably consists of the molecular complex with the octahedral structure of the copper-halide core. This assumption was supported by the quantum chemical study of the monomeric unit of the polymer. The recrystallization of the new solid phase from acetone or DMF returns the crystalline sample of the polymer. Therefore, the polymer can be considered to be a sensor for detecting the overheating of materials over 138 °C.

## Data Availability

Cambridge Crystallographic Data Centre #2225852.

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
