# Peer review of "Green Emissive Copper(I) Coordination Polymer Supported by the Diethylpyridylphosphine Ligand as a Luminescent Sensor for Overheating Processes"

_molecules, 2023, doi:10.3390/molecules28020706_

Round 1
Reviewer 1 Report
The reviewed manuscript describes a 1D coordination polymer (CP) assessed by treating CuI with diethylpyridylphosphine under mild conditions. The CP was structurally and spectrally characterized, as well as investigated in terms of solid-state luminescence. Moreover, DFT and TD-DFT computations were performed on a model fragment of the CP to understand its electronic structure and the nature of the electric transitions on the absorption spectrum. Overall, this work is well-performed, and it can be of interest to the readership of this journal. Thus, I recommend acceptance of this manuscript after addressing the following minor concerns:
1. Because the CP 2 does not retain 1D structure in solution, it is more relevant to compare the TD-DFT simulated pattern with the absorption spectrum recorded for the polycrystalline sample.
2. Authors proposed a phosphorescent origin of the emission because of “The large Stokes shift (ca. 8000 cm–1) and the microsecond domain of lifetimes”. Please note that the microsecond emission decays are also typical for TADF-emitting CuI-organic compounds. Thus, without a temperature-dependent photophysical study, it is difficult to clearly conclude the emission origin at ambient temperature.
3. The sentence “The solid-state quantum yield of complex 2 luminescence (20 %)…” should be better rephrased as follows: “The luminescence quantum yield of solid complex 2 (20 %)…”.
4. I didn't find details of DFT and TDDFT calculations. Please add these data in the MS or SI.
5. Finally, I recommend kindly extending the reference list over the recent relevant works on CuI-based emitters, e.g. DOI: 10.1039/C8QI01302K, 10.1039/C3DT53040J, 10.1002/anie.202103037, 10.1039/D1DT00826A, 10.1039/c9sc00970a, 10.1021/jacs.9b13772, 10.1002/adfm.201705593.
Author Response
Dear reviewer,
We would like to acknowledge you for the high evaluation of the manuscript and useful comments, which allowed to improve the quality of the manuscript. We have made several corrections according to your comments. Please find the point-by-point answers to your comments and questions below.
On the behalf on all authors,
Dr. Igor Strelnik
Comment 1. Because the CP 2 does not retain 1D structure in solution, it is more relevant to compare the TD-DFT simulated pattern with the absorption spectrum recorded for the polycrystalline sample.
Answer. The measure of solid-state UV/vis-absorbance is not a routine method and we don’t have opportunity to register the spectra in the solid state. At the same time, we suppose the obtained data are enough to conclude about structures, photophysical properties and its origins. Indeed, for the comparison of the simulated and experimental UV/vis-absorbance spectra, data registered for the solutions have been used. However, the vertical transitions were calculated for gas-phase optimized structures. Obtained by these way geometries are expected to be closer to the existing in the solutions than to the packed in the crystal. Thus, simulated by this way spectra should be compared with the solution experiment. The analysis of the experimental and simulated UV/vis-spectra allow to obtain the possible models for further study of the excited states. It should be noted, that the emission and excitation spectra have been measured for the solid-state samples of 2 and 2a. The solid-state excitation wavelength differs from the solution absorbance band wavelength, as it often observed for transition metal complexes. Solid-state emission spectra have been compared with the TDDFT calculated T1-S0 transitions of suggested models . The TDDFT analysis showed the same tendency in the predicted emission wavelengths as in the experimentally observed, but with other energy differences probably due the difference in the “phases” of the calculations and experiment.
Comment 2. Authors proposed a phosphorescent origin of the emission because of “The large Stokes shift (ca. 8000 cm–1) and the microsecond domain of lifetimes”. Please note that the microsecond emission decays are also typical for TADF-emitting CuI-organic compounds. Thus, without a temperature-dependent photophysical study, it is difficult to clearly conclude the emission origin at ambient temperature.
Answer. According to our experience and to the literature data, TADF is characteristic for mono- and dinuclear copper(I) complexes, whereas L2Cu4X4 or L4Cu4X4 clusters exhibit phosphorescence. The T1-S0 transitions are well studied in literature, and analysis of the latter allows to make conclusions about the origins of observed emission. Thus, we assign observed emission to the phosphorescence. Unlikely, we don’t have a technical opportunity to study temperature-dependent photophysical properties of complexes.
Comment 3. The sentence “The solid-state quantum yield of complex 2 luminescence (20 %)…” should be better rephrased as follows: “The luminescence quantum yield of solid complex 2 (20 %)…”.
Answer. The sentence has been revised:
“The luminescence quantum yield of complex 2 in solid state (20 %) is rather high in comparison with that of the other representatives of polymeric copper(I) complexes”
Comment 4. I didn't find details of DFT and TDDFT calculations. Please add these data in the MS or SI.
Answer. Details of DFT and TDDFT calculations have been added to the “Materials and methods”
“Computational Methods. Quantum chemical calculations were performed with the Gaussian 16 [53] suite of programs. The ground/excited-state structures were optimized with the use of hybrid PBE0 functional [54] and the Ahlrichs’ triple-z def-TZVP AO basis set [55]. In all geometry optimizations, the D3 approach [56] to describe the London dispersion interactions together with the Becke–Johnson damping function [57-59] were employed as implemented in the Gaussian 16 program. Time-Dependent Density Functional Response Theory (TDDFT) with the use of CAM-B3LYP [60] has been employed to compute the vertical excitation energies (i.e., absorption wavelengths) and oscillator strengths for the ground-state optimized geometries in the gas phase, 50 lowest singlet excited states were taken into account. The vertical T1-S0 transition energies were also computed within the TDDFT approach at the CAM-B3LYP/def-TZVP level at the triplet geometries optimized by UPBE0–D3(BJ)/def-TZVP level of theory. Calculated spectra were shifted by –0.24 eV.”
Comment 5. Finally, I recommend kindly extending the reference list over the recent relevant works on CuI-based emitters, e.g. DOI: 10.1039/C8QI01302K, 10.1039/C3DT53040J, 10.1002/anie.202103037, 10.1039/D1DT00826A, 10.1039/c9sc00970a, 10.1021/jacs.9b13772, 10.1002/adfm.201705593.
Answer. We have added abovementioned references
Reviewer 2 Report
1. The authors have referred such descriptions :”..Commonly the copper(I) halides form di- or tetranuclear clusters..”, I think the author could draw a scheme on the different formation of the tetranuclear clusters/unit. See this ref. Colloid. Surface A., 656(2023)130475.
2. The marked refs in the main text is not uniform, please revise it.
3. “The reaction of phosphine 1 with copper(I) iodide in the 1:2 metal-to-ligand ratio in dichloromethane (Scheme 2) leads to the formation of a yellowish crystalline precipitate, which represents copper(I) polymer according to the X-ray diffraction (XRD) study. How about the 1:1 metal-to-ligand ratio in dichloromethane, could you get the product?
4. Please discuss the weak interaction and packing scheme for its structural feature of this polymer
5. Please improve the quality of Fig. 5c.
6. “The predicted UV/Vis spectrum for the monomeric model is slightly blue-shifted compared to the one predicted for a dimer (Figure 5c) and better matches the experimental curve suggesting the presence of the monomeric units in the solution rather than polymer structure”. I suggest the author could compare similar documents and highlight them, such as Coord. Chem. Rev., 445(2021) 214074 and Coord. Chem. Rev. 2020, 406:213145
7. Some typesetting errors should be revised, such as”The polymer chain is 99 formed along the shortest axis 0a of the unit cell (P21/n).”
Author Response
Dear reviewer,
We would like to acknowledge you for the high evaluation of the manuscript and useful comments, which allowed to improve the quality of the manuscript. We have made several corrections according to your comments. Please find the point-by-point answers to your comments and questions below.
On the behalf on all authors,
Dr. Igor Strelnik
Comment 1. The authors have referred such descriptions :”..Commonly the copper(I) halides form di- or tetranuclear clusters..”, I think the author could draw a scheme on the different formation of the tetranuclear clusters/unit. See this ref. Colloid. Surface A., 656(2023)130475.
Answer. We have added figure 1 to the introduction.
Figure 1. Schematic representation of three types of Cu4X4 clusters
Comment 2. The marked refs in the main text is not uniform, please revise it.
Answer. We revised and corrected all references mentioned in the manuscript.
Comment 3. “The reaction of phosphine 1 with copper(I) iodide in the 1:2 metal-to-ligand ratio in dichloromethane (Scheme 2) leads to the formation of a yellowish crystalline precipitate, which represents copper(I) polymer according to the X-ray diffraction (XRD) study. How about the 1:1 metal-to-ligand ratio in dichloromethane, could you get the product?
Answer. We tried to isolate the product from the synthesis with 1:1 metal-to-ligand ratio, unlikely we didn’t succeed in its characterization due its low air-stability. According to our experience this product is the dimeric complex of L2Cu2I2 composition, but not a tetrameric cluster with L4Cu4X4 composition.
Comment 4. Please discuss the weak interaction and packing scheme for its structural feature of this polymer
Answer. We have added following discussion:
«Further, the chains in the crystal assemble to a 1D layer parallel to 0ac by intermolecular nonclassical hydrogen bonds C(sp2)–H···I and π···π-interactions between neighboring pyridyl moieties. The layers in turn stack along the longest axis 0b by dispersion interactions forming a rather loose layer-by-layer structure of the crystal with a packing index of 68.2 %.».
Comment 5. Please improve the quality of Fig. 5c.
Answer. The quality of 5c have been improved
Comment 6. “The predicted UV/Vis spectrum for the monomeric model is slightly blue-shifted compared to the one predicted for a dimer (Figure 5c) and better matches the experimental curve suggesting the presence of the monomeric units in the solution rather than polymer structure”. I suggest the author could compare similar documents and highlight them, such as Coord. Chem. Rev., 445(2021) 214074 and Coord. Chem. Rev. 2020, 406:213145
Answer. The references have been cited
Comment 7. Some typesetting errors should be revised, such as ”The polymer chain is 99 formed along the shortest axis 0a of the unit cell (P21/n).”
Answer. The text has been revised.
